# Prevalence and determinants of caesarean section in private and public health facilities in underserved South Asian communities: cross-sectional analysis of data from Bangladesh, India and Nepal

Melissa Neuman,[1] Glyn Alcock,[1] Kishwar Azad,[2] Abdul Kuddus,[2] David Osrin,[1] Neena Shah More,[3] Nirmala Nair,[4] Prasanta Tripathy,[4] Catherine Sikorski,[1] Naomi Saville,[1] Aman Sen,[5] Tim Colbourn,[1] Tanja A J Houweling,[6] Nadine Seward,[1] Dharma S Manandhar,[5] Bhim P Shrestha,[5] Anthony Costello,[1] Audrey Prost[1]

▶ Prepublication history and additional material is available. To view please visit the journal (http://dx.doi.org/10.1136/bmjopen-2014-005982).

For numbered affiliations see end of article.

**Correspondence to**
Dr Audrey Prost;
Audrey.prost@ucl.ac.uk

## ABSTRACT

**Objectives:** To describe the prevalence and determinants of births by caesarean section in private and public health facilities in underserved communities in South Asia.

**Design:** Cross-sectional study.

**Setting:** 81 community-based geographical clusters in four locations in Bangladesh, India and Nepal (three rural, one urban).

**Participants:** 45 327 births occurring in the study areas between 2005 and 2012.

**Outcome measures:** Proportion of caesarean section deliveries by location and type of facility; determinants of caesarean section delivery by location.

**Results:** Institutional delivery rates varied widely between settings, from 21% in rural India to 90% in urban India. The proportion of private and charitable facility births delivered by caesarean section was 73% in Bangladesh, 30% in rural Nepal, 18% in urban India and 5% in rural India. The odds of caesarean section were greater in private and charitable health facilities than in public facilities in three of four study locations, even when adjusted for pregnancy and delivery characteristics, maternal characteristics and year of delivery (Bangladesh: adjusted OR (AOR) 5.91, 95% CI 5.15 to 6.78; Nepal: AOR 2.37, 95% CI 1.62 to 3.44; urban India: AOR 1.22, 95% CI 1.09 to 1.38). We found that highly educated women were particularly likely to deliver by caesarean in private facilities in urban India (AOR 2.10; 95% CI 1.61 to 2.75) and also in rural Bangladesh (AOR 11.09, 95% CI 6.28 to 19.57).

**Conclusions:** Our results lend support to the hypothesis that increased caesarean section rates in these South Asian countries may be driven in part by the private sector. They also suggest that preferences for caesarean delivery may be higher among highly educated women, and that individual-level and provider-level

### Strengths and limitations of this study

- This study had a large sample size (>45 000 births) and focused on socioeconomically disadvantaged communities, which are a priority for public health interventions in South Asia.
- Our data were not nationally representative, which limits the generalisability of our findings.
- We did not have information on some known predictors of caesarean section, which would have enhanced the completeness of our determinants analysis.

factors interact in driving caesarean rates higher. Rates of caesarean section in the private sector, and their maternal and neonatal health outcomes, require close monitoring.

## INTRODUCTION

Access to comprehensive emergency obstetric care, including caesarean section, is key to preventing the estimated 287 000 maternal and 2.9 million neonatal deaths that occur worldwide every year.[1] [2] Although debate continues about how to quantify the need for life-saving obstetric surgery, a 1985 WHO report suggested that the optimal population range for caesarean section rates is between 5% and 15%, and this endures as a reference.[3] [4] Caesarean section rates are increasing worldwide, albeit unequally: a recent analysis of Demographic and Health Survey (DHS) data in 26 South Asian and sub-Saharan African

countries found that rates were highest among the 'urban rich' in all countries, and lowest among the 'rural poor' in 18.[5] In all countries, fewer than 5% of mothers in the poorest wealth quintile delivered by caesarean.[6]

Caesarean sections conducted without clinical need can have adverse consequences for mothers and children. A 2008 WHO survey of 373 facilities across 24 countries found that unnecessary caesareans were associated with an increased risk of maternal mortality and serious outcomes for mothers and newborn infants, compared with spontaneous vaginal delivery.[7] Recent ecological analyses also highlighted strong associations between caesarean delivery and increased neonatal mortality in countries with low and medium caesarean section rates.[8] Unnecessary caesareans lead to considerable costs for families and health systems: an estimated 6.2 million unnecessary procedures were performed in 2008, costing approximately US$2.32 billion.[9]

Several South Asian countries have recorded substantial increases in caesarean section rates over the past decade. In Bangladesh, rates rose from 2% (2000) to 17% (2011); in India, from 3% (1992) to 11% (2006); and in Nepal, from 1% (2000) to 5% (2011).[10–15] Several studies in these countries have raised concerns about high caesarean rates in private facilities, and a recent DHS analysis speculated that national increases in caesarean section rates in South Asian countries could be driven in part by higher rates among deliveries in private sector facilities.[6 16–18] The literature from other settings indicates that increases in caesarean sections are shaped by supply and demand pressures: providers often have financial incentives to intervene surgically, and women of higher socioeconomic status are also more likely to opt for caesareans.[19 20]

There are few large, recent community-based studies from South Asia quantifying differences in caesarean section rates between public and private facilities. Such studies are necessary in order to examine whether increases in caesareans in these settings are indeed likely to be driven by the private sector, demand from wealthier and more educated mothers, or a combination of the two. We conducted a cross-sectional analysis of data from Bangladesh, India and Nepal to explore the prevalence and determinants of caesarean section delivery by type of facility and maternal characteristics.

## METHODS
### Study populations
We used data collected through vital events surveillance systems established during four cluster-randomised controlled trials (cRCTs) conducted between 2005 and 2011. The trials were conducted with communities that can be considered socioeconomically disadvantaged: in Bangladesh and Nepal, they took place in four rural, underserved districts (Bogra, Maulvibazaar and Faridpur in Bangladesh, and the Terai district of Dhanusha, Nepal); in rural India, most participants were from

Scheduled Tribes in two states of eastern India (Jharkhand and Odisha); in urban India, data came from informal settlements (slums) in Mumbai. Table 1 describes the characteristics of each study and its population, including the background maternal mortality ratio and types of facilities present in the study areas. The original trials were designed to evaluate the impact of participatory women's groups on maternal and neonatal health outcomes.[21–25] We used data from the control areas of these trials only, because the women's group intervention led to changes in mortality and practices in several locations.[26]

### Health system contexts
All three study countries have experienced substantial increases in institutional delivery rates over the past two decades: births in health facilities increased from 4% (1993) to 29% (2011) in Bangladesh; from 26% (1992–1993) to 47% (2007–2008) in India and from 8% (1996) to 35% (2011) in Nepal.[10–15] All three countries have implemented incentive schemes to promote institutional delivery, though these have varying coverage. In 2010, Bangladesh's pilot maternity voucher scheme reached an estimated 10.4 million people across 31 subdistricts, around 7% of the country's population.[27 28] Mothers receive a cash incentive for antenatal care and delivery in a public or private facility, or at home with a skilled birth attendant. Government as well as private facility staff also receive cash incentives, including 3000 Bangladeshi Taka (US$38.5) for a caesarean section and 300 Tk for a normal delivery.[28] This maternity incentive scheme was operational in two of the three districts (Faridpur and Maulvibazaar) covered by our study. All three districts had public facilities including District Hospitals, Maternal and Child Welfare Centres, and Upazilla Health Complexes. Private facilities included a number of small-to-medium size clinics, Bangladesh Rural Advancement Committee (BRAC; non-governmental organisation, NGO) facilities and larger private hospitals.

In India, the *Janani Suraksha Yojana* (JSY) maternity incentive scheme entitles women in rural areas of high focus states, including those in this study, to ₹1400 (US$22.4) after delivering in a government or accredited private health facility. Local community health volunteers called Accredited Social Health Activists (ASHAs) also receive ₹600 (US$9.6) for identifying pregnant women and helping them get to a health facility.[29] Although the rural Indian data included in this study were collected between 2005 and 2008, the JSY was only operational in the study areas from 2008 onwards and its impact is unlikely to be reflected here. The Indian urban area included in our study spanned informal settlements (slums) with a wealth of public and private providers.[30] Mothers with Below Poverty Line cards in such areas are eligible for JSY and can receive ₹500 towards the costs of delivering in a health facility.[31] Nepal began a safe delivery incentive scheme in 2005 and free deliveries have been available in government facilities since

**Table 1**  Characteristics of studies and populations

| Study (country) | Bangladesh (rural) | India (rural) | Nepal (rural) | India (urban) |
|---|---|---|---|---|
| Location | Three districts: Bogra, Maulvibazaar and Faridpur | Three districts of Jharkhand and Odisha: Keonjhar, West Singhbhum and Saraikela | Dhanusha district (Terai) | Mumbai slums |
| Period | 2005–2011 | 2005–2008 | 2008–2011 | 2006–2009 |
| Estimated population | 532 900 | 114 000 | 240 000 | 283 000 |
| Cluster characteristics | Villages making up a union | 8–10 villages with residents classified as Scheduled Tribe or Other Backward Class | Village Development Committee | Slum areas in six municipal wards of Mumbai |
| Method of cluster identification | Purposive sampling of three districts and clusters within districts | Purposive sampling of three districts and clusters within districts | Random sampling of 60 clusters from a list of 79 suitable clusters in one district | 92 clusters in six municipal wards identified using municipal documents, surveys, discussions with key informants, and site visits. Random selection of 48 clusters for randomised allocation |
| Clusters, n | 9 | 18 | 30 | 24 |
| Cluster and individual follow-up | All clusters followed up Interviews completed after 82% of identified births in control areas in Phase 1, and 99% of births in Phase 2 | All clusters followed up Interviews completed after 98% of identified births | All clusters followed up | All clusters followed up Interviews completed after 83% of identified births |
| Maternal mortality ratio | 254.3 | 668.1 | Unknown | 206.2 |
| Health facilities available in control areas | Public facilities: District Hospitals; Maternal and Child Welfare Centres; Upazilla Health Complexes. Private facilities: small-to-medium size clinics; BRAC (NGO) facilities where deliveries do not take place; larger private hospitals with and without CEmOC facilities | Public facilities: District Hospitals; PHCs in which deliveries can notionally take place but that are not usually equipped for CEmOC; CHCs acting as referral centres for PHCs, covering a population of around 80 000 with EmOC facilities; district hospitals. Private and charitable facilities: medium-sized missionary hospitals with EmOC facilities | Public facilities: three Primary Health Care Centres, three Health Posts and 24 Sub-Health Posts, none of which are equipped for CEmOC. These health facilities refer to the public Zonal Tertiary Hospital and various private providers in the district headquarters and nearby medical college, which have facilities for caesarean sections | Public facilities: municipal tertiary hospitals, general hospitals and maternity homes. Private facilities: specialty hospitals, general hospitals and maternity homes |

BRAC, Bangladesh Rural Advancement Committee; CEmOC, comprehensive emergency obstetric care; CHC, Community Health Centre; NGO, non-governmental organisation; PHC, Primary Health Centre.

2009 through the *Aama Surakshya* programme.[32] Dhanusha district, from where data for this study came, has one zonal tertiary hospital and a private medical college hospital equipped for caesarean sections, and a variety of small public and private health facilities without comprehensive obstetric care.

## Data collection

All data were collected using surveillance systems to monitor births and deaths prospectively.[21–25] In all study locations, a female community-based key informant reported births and deaths in her area, which covered a population ranging from 250 to 350. A trained interviewer then verified these reports and paid the informant an incentive for each correct identification. In Bangladesh, and rural and urban India, the interviewer administered a structured questionnaire to all eligible mothers around 6 weeks after delivery; in Nepal, all births in the study area were registered, and interviews were conducted on all births in small clusters and on a random sample of 10 births per month in the larger clusters. In each study location, mothers were interviewed using a questionnaire to collect information about events in the antenatal, delivery and postnatal periods. Participants were women of reproductive age (15–49) who delivered in the study areas during the data collection periods, and who consented to be interviewed 6 weeks after delivery, as well as their infants.

## Study sample

The initial sample for this analysis included 46 393 births in Mumbai, rural India, rural Bangladesh, or rural Nepal, of whom 17 565 (38%) were delivered in healthcare facilities. We excluded the 2348 births occurring outside Mumbai, because we collected limited information on delivery location for these. Additionally, 223 deliveries had missing data on form of delivery (caesarean or vaginal), 602 had missing or implausible values for mother's age and 98 were missing information on other maternal characteristics (educational attainment, household assets, number of prior pregnancies, number of antenatal visits, severe problems in pregnancy or delivery); 143 births had missing or incomplete information on place of delivery. The final analytic sample included 45 327 births (97.7% of the initial sample).

## Measures used

The primary outcome in these analyses was caesarean section delivery, identified by self-report from the mother or another household member around 6 weeks after giving birth. The main covariate of interest was the type of delivery facility, coded as public (eg, district hospitals), private (individual private clinics or hospitals), or NGO/charitable (eg, Christian missionary hospitals in rural India; BRAC clinics in Bangladesh). We grouped private and NGO/charitable facilities together due to the small number of births in NGO/charitable facilities (6% of institutional deliveries in Bangladesh, 7% in rural India, and

0% in urban India and rural Nepal; data on request). Additional covariates in the models included measures of the characteristics of the pregnancy and delivery, maternal sociodemographic characteristics and the location in which the delivery occurred. We created an indicator variable to identify women experiencing serious problems during pregnancy or delivery, with women reporting symptoms of preeclampsia (blurred vision or swelling of face and hands), symptoms of eclampsia (fits or seizures during pregnancy or delivery), haemorrhage during delivery, or labour lasting more than 24 h considered to have serious complications. Other characteristics of the pregnancy and delivery considered in the analysis included full utilisation of antenatal care, number of prior pregnancies and multiple pregnancy. Full utilisation of antenatal care was defined as four or more visits to antenatal care, with at least one visit to a skilled provider. Number of pregnancies was entered as an ordered categorical variable, with categories of one (first), two, three, or four or more pregnancies.

Maternal sociodemographic characteristics included in the models were: mother's age at delivery, her educational attainment and household assets. Mother's age was entered into the model as a categorical measure in 10-year groups. Educational attainment was entered as a categorical variable using the following categories: no formal education, primary education, secondary education, or bachelor degree or higher. To develop an asset index, we used polychoric factor analysis on data on common assets and amenities found in the mother's household, and grouped the resulting factor scores into quartiles.[33] Assets and amenities included electricity, radio or cassette player, electric fan (Bangladesh and India only), television, refrigerator (Bangladesh and India only), telephone (Bangladesh and Nepal only), generator (India only) and bicycle. All models were additionally adjusted for location (Bangladesh, India and Nepal), and year of interview in 3-year groups. The data were collected in a stratified, cluster-sampled survey, and we accounted for survey design in the analysis using a fixed effect for stratum and a random effect for cluster.

## Statistical analysis

We used frequencies to describe caesarean section rates by delivery location at each site. We used the Generalized Linear Latent And Mixed Models procedure in Stata V.13.1, with adaptive quadrature for binary outcomes, to estimate the crude association between type of delivery facility and caesarean section.[34 35] We identified other maternal, pregnancy and delivery characteristics potentially associated with caesarean section using the existing literature, especially studies resulting from the WHO multicountry surveys, and entered these in adjusted models to explore how they modified the association between type of delivery facility and caesarean section, and their individual association with caesarean section. Some South Asian and Latin American studies have detected a strong association

between maternal education and caesarean delivery.[7][36][37] We fitted models including indicator variables for each group of mothers by education and type of delivery facility to explore differences in the strength of association between caesarean delivery and private facility by mother's education. To account for the sampling procedure used in rural Nepal, models were adjusted using pweights (probability of selection within cluster); these weights were rescaled to reflect the total number of institutional deliveries.

## RESULTS

Table 2 describes the characteristics of institutional births and caesarean deliveries by location. We analysed data for 45 327 births: 21 560 in rural Bangladesh, 8541 in rural India, 10 236 in urban India and 4931 in rural Nepal. The proportion of women delivering in health facilities varied widely between locations, from 90% in urban India to 21% in rural India. There were also large variations in the proportion of women delivering in private/charitable facilities rather than public facilities, with the highest (77%) in rural India and the lowest (10%) in rural Nepal. The proportion of women giving birth by caesarean section in private rather than public facilities also varied widely between settings. In Bangladesh, only 21% of women delivered in a health facility, around half of them in the private/charitable sector, but 73% of private facility births were by caesarean section. In rural Nepal, 30% of the 162 private facility births were by caesarean section. In informal settlements of Mumbai, 15% of public facility deliveries were caesareans, compared with 18% of private facility deliveries. In rural India, caesarean sections were more commonly performed in public health facilities than private or charitable facilities (15% vs 5%). Online Supplementary table S1 describes the characteristics of institutional deliveries (caesarean or non-caesarean) according to type of delivery facility (private or public), maternal socioeconomic and sociodemographic characteristics, pregnancy and delivery characteristics, and year of delivery.

Table 3 shows crude and adjusted measures of association between type of delivery facility (private or public) and caesarean section for each location. Delivering in a private health facility was associated with increased odds of caesarean section in all but one location (rural India). The relative odds of a caesarean in a private facility were greatest in Bangladesh (OR 6.82, 95% CI 5.96 to 7.81), followed by Nepal (OR 2.42, 95% CI 1.48 to 3.94) and urban India (OR 1.36, 95% CI 1.21 to 1.52). These associations persisted and were only mildly attenuated when adjusted for maternal characteristics, pregnancy and delivery characteristics, and year of delivery (Bangladesh: adjusted OR (AOR) 5.91, 95% CI 5.15 to 6.78; Nepal: AOR 2.37, 95% CI 1.62 to 3.44; urban India: AOR 1.22, 95% CI 1.09 to 1.38).

**Table 2** Characteristics of institutional births and caesarean deliveries by location

| | Births (n) | Institutional births | | | | | | Caesarean section births | | | |
| | | All | | Public facility | | Private/charitable facility | | Public facility | | Private/charitable facility | |
| | | (n) | (% all births) | (n) | (% institutional births) | (n) | (% institutional births) | (n) | (% deliveries in public facilities) | (n) | (% deliveries in private facilities) |
| Bangladesh (rural) | 21 560 | 4592 | 21 | 2053 | 45 | 2539 | 55 | 589 | 29 | 1852 | 73 |
| India (rural) | 8541 | 1816 | 21 | 421 | 23 | 1395 | 77 | 64 | 15 | 66 | 5 |
| India (urban) | 10 236 | 9259 | 90 | 5489 | 59 | 3770 | 41 | 811 | 15 | 697 | 18 |
| Nepal (rural)* | 4931 | 1586 | 32 | 1424 | 90 | 162 | 10 | 201 | 14 | 48 | 30 |

*Nepal numbers weighted using women's probability of selection within each cluster, scaled to total number of institutional births (unweighted Nepal totals: 4990 births and 1586 institutional births).

**Table 3** Mutually adjusted associations of type of facility of delivery and other selected determinants with caesarean section delivery, by location

| | BD (rural) ORs (95% CIs) | | IN (rural) ORs (95% CIs) | | IN (urban) ORs (95% CIs) | | NP (rural)* ORs (95% CIs) | |
|---|---|---|---|---|---|---|---|---|
| | Crude | Adjusted | Crude | Adjusted | Crude | Adjusted | Crude | Adjusted |
| **Health facility characteristics** | | | | | | | | |
| Public health facility (ref) | | | | | | | | |
| Private health facility | 6.82 (5.96 to 7.81) | 5.91 (5.15 to 6.78) | 0.39 (0.25 to 0.61) | 0.46 (0.29 to 0.73) | 1.36 (1.21 to 1.52) | 1.22 (1.09 to 1.38) | 2.42 (1.48 to 3.94) | 2.37 (1.62 to 3.44) |
| **Pregnancy and delivery characteristics** | | | | | | | | |
| 4+ antenatal care visits (fewer or none=ref)† | | 1.46 (1.26 to 1.69) | | 1.49 (0.96 to 2.32) | | 1.06 (0.93 to 1.21) | | 1.92 (1.43 to 2.58) |
| **Birth order** | | | | | | | | |
| 1 (ref) | | | | | | | | |
| 2 | | 0.95 (0.80 to 1.13) | | 1.25 (0.80 to 1.95) | | 0.65 (0.56 to 0.75) | | 1.41 (1.02 to 1.94) |
| 3 | | 1.19 (0.94 to 1.52) | | 0.36 (0.15 to 0.87) | | 0.60 (0.50 to 0.71) | | 0.58 (0.37 to 0.92) |
| 4+ | | 1.17 (0.90 to 1.53) | | 0.36 (0.15 to 0.85) | | 0.39 (0.32 to 0.48) | | 1.00 (0.54 to 1.84) |
| Serious complications in pregnancy/delivery‡ (no complications=ref) | | 0.87 (0.76 to 1.00) | | 1.77 (1.17 to 2.67) | | 1.71 (1.39 to 2.11) | | 4.87 (2.51 to 9.47) |
| Multiple birth (ref=single) | | 0.93 (0.65 to 1.32) | | 1.57 (0.52 to 4.75) | | 3.01 (2.14 to 4.23) | | 3.42 (1.77 to 6.61) |
| **Maternal characteristics** | | | | | | | | |
| **Age (years)** | | | | | | | | |
| 15–24 (ref) | | | | | | | | |
| 25–34 | | 1.12 (0.94 to 1.34) | | 1.81 (1.14 to 2.86) | | 1.45 (1.27 to 1.66) | | 1.52 (0.63 to 1.94) |
| 35+ | | 1.10 (0.77 to 1.58) | | 2.11 (0.63to 7.07) | | 1.79 (1.27 to 2.53) | | 1.55 (0.55 to 1.47) |
| **Maternal education§** | | | | | | | | |
| No formal education (ref) | | | | | | | | |
| Primary education | | 1.07 (0.83 to 1.38) | | 0.98 (0.39 to 2.47) | | 1.13 (0.85 to 1.49) | | 0.90 (0.59 to 1.39) |
| Secondary education | | 1.44 (1.13 to 1.84) | | 1.11 (0.68 to 1.80) | | 1.22 (1.04 to 1.42) | | 1.37 (0.97 to 1.94) |
| Bachelor degree or higher | | 2.44 (1.52 to 3.92) | | 1.20 (0.43 to 3.31) | | 1.62 (1.30 to 2.02) | | – |
| **Household wealth quintile** | | | | | | | | |
| 1st (poorest, ref) | | | | | | | | |
| 2nd | | 1.31 (1.06 to 1.63) | | 1.61 (0.69 to 3.75) | | 1.14 (0.77 to 1.68) | | 1.11 (0.63 to 1.94) |
| 3rd | | 1.41 (1.10 to 1.82) | | 1.36 (0.58 to 3.21) | | 1.13 (0.95 to 1.33) | | 0.90 (0.55 to 1.47) |
| 4th (wealthiest) | | 1.36 (1.09 to 1.70) | | 2.16 (0.87 to 5.33) | | 1.50 (1.27 to 1.78) | | 0.99 (0.52 to 1.90) |
| **Year of delivery** | | | | | | | | |
| 2004–2006 (ref: BD, IN) | | | | | | | | – |
| 2007–2009 (ref: NP rural) | | 0.90 (0.74 to 1.10) | | 0.90 (0.61 to 1.33) | | 1.14 (1.09 to 1.38) | | – |
| 2010–2012 | | 1.15 (0.94 to 1.41) | | – | | – | | 1.29 (0.91 to 1.82) |
| N | 4592 | 4592 | 1816 | 1816 | 9259 | 9259 | 1586 | 1586 |

All analyses additionally adjusted for survey design using fixed effect of stratum and random effect of cluster.
*Nepal numbers weighted using women's probability of selection within each cluster, scaled to total number of institutional births.
†At least one visit with skilled provider.
‡Includes: symptoms of eclampsia (fits, seizures, convulsions, or unconsciousness during pregnancy or delivery); reduced or no fetal movement; labour lasting more than 24 h.
§For the Nepal data, two respondents with bachelor degrees (2 respondent total, 1 delivering in institution) were combined with respondents with secondary education.
BD, Bangladesh; IN, India; NP, Nepal.

Women who had four or more antenatal check-ups were significantly more likely to have a caesarean delivery in rural Bangladesh and Nepal, with positive but non-significant trends in all other locations. Parity was not associated with caesarean delivery in any location. Having a serious health complication during pregnancy and delivery was associated with caesarean delivery in all locations except rural Bangladesh, where we observed a negative association (AOR 0.87, 95% CI 0.76 to 1.00). Multiple birth was associated with increased odds of caesarean section only in urban India and rural Nepal. Higher maternal age was only associated with increased odds of caesarean section in urban India. Maternal education was only associated with increased odds of caesarean delivery in rural Bangladesh and urban India, with mothers with secondary education or Bachelor degrees having higher odds (AOR 1.44, 95% CI 1.13 to 1.84 and AOR 2.44, 95% CI 1.52 to 3.92 for Bangladesh and AOR 1.22, 95% CI 1.04 to 1.42 and 1.62, 95% CI 1.30 to 2.02 for urban India, respectively). There were significant positive associations only in rural Bangladesh, with small increases in odds of caesarean section for unit increase in wealth quartile (AOR comparing wealthiest to poorest group in Bangladesh: 1.36, 95% CI 1.09 to 1.70), and in urban India, for the wealthiest quartile (AOR 1.50, 95% CI 1.27 to 1.78).

We found interactive associations between maternal education and private facility delivery in two of four sites, with including an interaction improving model fit in two of the four analyses (p=0.021 in rural Bangladesh and p<0.001 in urban India; table 4 and online supplementary table S2). In Bangladesh, there was a pronounced positive educational gradient in caesarean delivery within those who delivered in private facilities, as well as a positive association between private facility delivery and caesarean delivery. Women with bachelor degrees delivering in private facilities had 11 times greater odds of delivering by caesarean (AOR 11.09, 95% CI 6.28 to 19.57) than did women with no formal education delivering in public facilities. Women with no formal education delivering in private facilities had nearly seven times the odds of caesarean compared with the reference group (AOR 6.94, 95% CI 4.64 to 10.39). However, the interaction with educational attainment was negative, suggesting that women holding bachelor degrees are somewhat less likely than women with secondary degrees to deliver by caesarean (online supplementary table S2). A positive gradient was seen among women delivering in public facilities, with women with bachelor education having four times greater odds of caesarean delivery in public facilities than women with no formal education (AOR 4.55, 95% CI 2.22 to 9.33). There was also an educational gradient among women delivering in private facilities in urban India. Educated mothers delivering in private facilities were more likely to deliver by caesarean than were mothers in other groups (AOR 2.10; 95% CI 1.61 to 2.75). However, there was no apparent gradient by

**Table 4** Interactive associations between type of delivery facility, maternal education, and caesarean delivery, by location

| | Bangladesh (rural) | | India (rural) | | India (urban) | | Nepal (rural)* | |
|---|---|---|---|---|---|---|---|---|
| | (n) | AOR, 95% CI | (n) | AOR, 95% CI | (n) | AOR, 95% CI | (unweighted n) | AOR, 95% CI |
| Public facility—no maternal education (ref) | 202 | – | 109 | – | 1396 | – | 822 | – |
| Public facility—primary education | 448 | 1.07 (0.76 to 1.50) | 17 | 0.92 (0.18 to 4.77) | 357 | 0.88 (0.61 to 1.27) | 196 | 1.04 (0.65 to 1.65) |
| Public facility—secondary education | 202 | 1.63 (1.18 to 2.24) | 604 | 1.09 (0.52 to 2.26) | 920 | 1.1 (0.91 to 1.33) | 413 | 1.45 (1.04 to 2.02) |
| Public facility—bachelor degree | 448 | 4.58 (2.24 to 9.38) | 100 | 1.55 (0.41 to 5.89) | 180 | 1.06 (0.76 to 1.47) | – | – |
| Private facility—no education | 332 | 7.24 (4.82 to 10.87) | 109 | 0.46 (0.21 to 1.01) | 1396 | 0.91 (0.70 to 1.19) | 100 | 2.83 (1.73 to 4.61) |
| Private facility—primary education | 578 | 7.58 (5.36 to 10.71) | 17 | 0.47 (0.14 to 1.58) | 357 | 1.52 (0.99 to 2.33) | 16 | 0.86 (0.14 to 5.21) |
| Private facility—secondary education | 1102 | 9.03 (6.57 to 12.41) | 274 | 0.52 (0.24 to 1.12) | 3382 | 1.3 (1.06 to 1.60) | 39 | 3.01 (1.71 to 5.30) |
| Private facility—bachelor degree | 41 | 11.24 (6.37 to 19.85) | 21 | 0.4 (0.08 to 2.00) | 354 | 2.1 (1.61 to 2.75) | – | – |
| N | 3353 | | 1251 | | 8342 | | 1586 | |
| p Value† | 0.021 | | 0.792 | | 0.001 | | 0.394 | |

All results adjusted for number of antenatal care visits, parity, medical indication for caesarean, multiple birth, maternal age (10-year group), household assets, stratum and cluster (random effect).
*Nepal numbers weighted using women's probability of selection within each cluster, scaled to total number of institutional births.
†p Value from −2 log-likelihood test comparing nested models with and without interaction terms.
AOR, adjusted OR.

educational attainment among women delivering in public facilities.

## DISCUSSION

Our analysis of data from over 45 000 births confirms the findings of other studies identifying differences in caesarean rates between public and private facilities, and suggests that, even in underserved areas in South Asia, caesareans without medical indication are of concern. In three of four locations, rates of caesarean section were higher in private/charitable facilities than in public facilities. The findings from Bangladesh are particularly noteworthy as they show much greater odds of caesarean section in private facilities, concurring with previous analyses.[37] This was also the only location where serious complications in pregnancy and delivery were not associated with caesarean delivery, suggesting that obstetric surgery was performed over and above clinical need.

Our findings confirm the results of earlier studies of the prevalence of caesarean delivery in South Asia, and indicate that high rates can be found in underserved rural areas. In India, a recent analysis of 2010–2011 Annual Health Survey (AHS) data from 284 districts in nine States, including Jharkhand, found that the median caesarean section rate in the private sector was 28%, compared with 5% in the public sector.[38] There appear to be strong financial incentives for surgical procedures in the private sector.[39] In a rural, largely indigenous part of eastern India, where more mothers delivered in private/charitable facilities than in public facilities, more caesarean sections were performed in the public sector. In this particular setting, our field experience suggests that only women with serious complications would go to a facility and have a caesarean section, especially as JSY was not yet available in the study areas at the time of data collection. Such women are likely to have experienced multiple referrals from either ill-equipped public or private facilities not wanting to take the risk of admission. It is, however, possible that preference for the private sector has changed in Jharkhand and Odisha since the advent of JSY, and further disaggregated analyses of AHS data would allow a more contemporary exploration of state-level variations in caesarean section rates between public and private sectors.

In Mumbai slums with a high uptake of institutional births and a 60/40 split in favour of the public sector for delivery care, we found no difference in caesarean section rates between the two sectors, which is somewhat surprising given the high rates of caesareans observed in the private sector in nine other states of India (excluding Maharashtra). At least two scenarios may be relevant in this setting. First, public sector hospitals are able to provide caesarean section, while smaller private maternity homes may not be able to. Second, the costs of caesarean section borne by private sector providers are higher in Mumbai, and families may be unwilling to bear these additional costs. This provides motivation for private providers not to provide caesareans, and makes it likely that women experiencing complications move into the public sector. The variety of facility options in an urban environment means that mothers are able to choose between different types of facilities, while in rural areas only one provider may be accessible.

Analyses using the most recent Bangladesh DHS found that, in 2011, three in five facility births were delivered by caesarean section. This reflects a historical trend: in 2001–2003, nearly half of deliveries in private facilities in Bangladesh were already by caesarean section.[18] Our data from rural, socioeconomically disadvantaged communities in three districts of Bangladesh confirm these population-level findings, but also suggest that high caesarean section rates in private facilities are not merely an issue for wealthy urban mothers. Although some research from Bangladesh suggests that mothers may have a preference for caesarean delivery because of fear of labour pain or a desire to select an auspicious date for the birth, other studies also highlight women's fears of caesarean section and their distrust of health providers who recommend them, mainly because of the high costs associated with the procedure.[37 40 41] A qualitative study involving 20 women who had experienced obstetric complications in Matlab in 2008–2009 found that most of the 14 women who had undergone caesareans had spent over 14 999 Tk (US$217) on the procedure, which was approximately one-third of gross domestic product per capita at the time.[41] The lack of association between caesareans and complications in pregnancy or delivery and multiple births, coupled with the high financial incentives given to providers for performing caesarean sections and the requirement for junior doctors to 'practise' their surgical skills, further suggest that obstetric surgery is being used over and above clinical need. A possible explanation for the significant interactive associations between maternal education and caesarean delivery in public and private facilities in Bangladesh is that well-educated women may be delivering in more expensive or highly rated institutions, which may in turn be more likely to perform caesarean sections for financial reasons and if they act as training centres for junior doctors.

In our sample of institutional deliveries from rural Nepal, 16% of facility births were by caesarean, which is higher than the national average: the 2011 Nepal DHS found an overall national caesarean section rate of 5%, with sections more commonly performed for births to highly educated mothers (13%) and mothers in the highest wealth quintile (14%).[26] It is possible that women with complications are more likely to deliver in facilities, and also that, as in Bangladesh, private providers are motivated by financial incentives to conduct caesareans more frequently than strictly necessary. Women who can afford to seek care in private facilities may also be more willing or able to pay for caesarean sections, and providers may conduct more of them to increase their income.

A 2010 analysis of DHS data examining the role of the private sector in maternity care in 16 countries found evidence of a trend towards privatisation in delivery care between the 1990s and mid-2000s, but with strong differences between countries, which might reflect the heterogeneous nature of this sector between and within countries.[42] This DHS study highlighted the need for more context-specific data on the nature of the private sector in low-income and middle-income countries, and its role in maternity care. Further research might focus on understanding the motivations and experiences of women undergoing caesarean sections in private facilities in South Asian settings, pathways for switching between public and private sectors in the event of obstetric complications, a more comprehensive tally of the financial incentives (official or non-official) that motivate private providers to carry out caesareans in each setting, and the consequences of increased caesarean sections in the private sector for maternal and neonatal health outcomes.

In our study, locations with higher prevalence of caesarean deliveries also showed a positive educational gradient even after adjusting for wealth measured by household assets. Moreover, there was an interactive association between education and type of facility, with highly educated women particularly likely to receive caesarean deliveries in private facilities in urban India. The literature exploring the determinants of caesarean delivery emphasises that multiple influences drive a woman's decision to deliver by caesarean. Profit considerations at the facility level may prompt some providers to urge women to receive unnecessary procedures, while women themselves may prefer caesareans for cultural reasons, fear of painful deliveries, or because they believe them to be safer.[36 43–45] Within the client–provider interaction, providers may be more likely to acquiesce to a request for caesarean from a highly educated woman; and such women may be more likely to accept advice from a provider.[19 46] Our findings lend support to the hypothesis that, while provider-level factors are probably partially responsible for the rapid increase in caesarean deliveries, it is also necessary to consider women's own preferences and decision-making processes and how they are shaped by social and cultural factors.

## Strengths and limitations

The strengths of the study were its large sample size and focus on underserved communities, which are a priority for public health interventions in South Asia. It had five main limitations. It was not a nationally representative study, and districts or clusters were sampled purposively from previous cRCTs. This limits the generalisability of our findings to geographical settings outside the study areas. The study was cross-sectional, and therefore only able to suggest associations rather than causal relationships. The data also did not include key predictors of caesarean section such as breech presentation, and information on whether the previous delivery was by caesarean, which limits the completeness of our analysis of

determinants. In addition, levels of serious complications in pregnancy and delivery varied considerably between locations and were often higher than expected, suggesting potential over-reporting and limited reliability as an indicator of complications. Finally, some variables had small denominators, and others had high levels of missing data (eg, maternal age in rural India).

## Conclusions

Our study found that delivering in a private health facility was associated with increased odds of caesarean section in three of four South Asian locations, and that the associations persisted after adjustments for maternal, pregnancy and delivery characteristics, and year of delivery. We also found significant interactive associations between maternal education and caesarean delivery in Bangladesh (private as well as public facilities) and urban India (private facilities only). These results lend support to the hypothesis that increased caesarean section rates in these three South Asian countries may in part be driven by the private sector, but also suggest that, in some settings such as Bangladesh and urban India, demand from more educated mothers may play a part. These findings call for greater, local understanding of the role of private providers in maternity care, together with careful examination of the consequences of increased caesarean sections in the private sector for maternal and neonatal health.

**Author affiliations**
[1]Institute for Global Health, University College London, London, UK
[2]Perinatal Care Project, Diabetic Association of Bangladesh, Dhaka, Bangladesh
[3]Society for Nutrition, Education and Health Action (SNEHA), Urban Health Centre, Maharashtra, India
[4]Ekjut, Jharkhand, India
[5]Mother and Infant Research Activities (MIRA), Kathmandu, Nepal
[6]Department of Public Health, Erasmus MC University Medical Center Rotterdam, Rotterdam, The Netherlands

**Contributors** MN and AP had the original idea for the study. MN designed the data analysis strategy in discussion with AP. MN, GA, AS, CS and NS prepared the data. MN performed the analysis and prepared the tables. NN, PT, NSM, DO, KA, DSM and AC are responsible for the trials. AP wrote the first draft of the paper and all authors read and commented on the draft, and also read and approved the final manuscript.

**Funding** MN and NS are funded through a Wellcome Trust Strategic Award (085417MA/Z/08/Z). In Bangladesh, the Perinatal Care Project trial (covering KA and AK) was registered with trial number ISRCTN54792066 and has received funding from the Big Lottery Fund (IS/2/010281409) and the Wellcome Trust (85417MA/Z/08/Z). In Eastern India, the Ekjut trial (covering NN and PT) was registered with trial number ISRCTN21817853 and funded by the Health Foundation (1748/3001). The Dhanusha (Nepal) trial (covering DSM, NS, BPS and CS) was registered with trial number ISRCTN87820538 and has received funding from: the UBS Optimus Foundation; USAID; DfID and the Wellcome Trust. The City Initiative for Newborn Health trial led by SNEHA (Mumbai, India) was registered with trial number ISRCTN96256793 and has received funding from the ICICI Foundation for Inclusive Growth and the Wellcome Trust (grant ref: 081052). GA, NSM and DO are currently funded by The Wellcome Trust (091561/Z/10/Z).

**Competing interests** None.

**Ethics approval** The ethics committee of the Diabetic Association of Bangladesh (Perinatal Care Project, Bangladesh Diabetes Somity, BADAS); an independent ethics committee in Jamshedpur, Jharkhand, India; the Nepal Health Research Council; the Municipal Corporation of Greater Mumbai and the Independent Ethics Committee for Research on Human Subjects (Mumbai, India); and the ethics committee of the Institute of Child Health, University College London.

**Provenance and peer review** Not commissioned; externally peer reviewed.

**Data sharing statement** Data used in the analyses for this study are available on request, subject to completion of a data sharing agreement. Please contact audrey.prost@ucl.ac.uk for further information.

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
