## [Reviewer comments · BMJ Open]

ARTICLE DETAILS

TITLE (PROVISIONAL)	Prevalence and determinants of caesarean section in private and public health facilities in underserved South Asian communities: cross-sectional analysis of data from Bangladesh, India, and Nepal
AUTHORS	Neuman, Melissa; Alcock, Glyn; Azad, Kishwar; Kuddus, Abdul; Osrin, David; Shah More, Neena; Nair, Nirmala; Tripathy, Prasanta; Sikorski, Catherine; Saville, Naomi; Sen, Aman; Colbourn, Timothy; Houweling, Tanja; Seward, Nadine; Manandhar, Dharma; Shrestha, Bhim; Costello, Anthony; Prost, Audrey

VERSION 1 - REVIEW

REVIEWER	Caroline Homer University of Technology Sydney Australia
REVIEW RETURNED	05-Jul-2014

GENERAL COMMENTS	Thank you for the opportunity to review this paper on the prevalence and determinates of Caesarean section in under served communities in South Asia. The authors have undertaken a secondary analysis of data from 4 trials and have analysed these separately and together for this paper. Private – public status were examined as well as age, education, wealth and complications of pregnancy and birth. For me, the findings were a combination of being profoundly depressing and highly concerning. Very high rates of CS were reported (up to 73% in private settings in Bangladesh) and financial and training incentives for CS were also a strong theme. The aims and objectives are clear and the methodology appropriate and clearly described. The authors have followed the STROBE format for reporting studies of this nature. I recommend publication of this important paper with a few of minor comments that I feel would strengthen it. The fact that this is a secondary analysis of data from previous trials needs to be included under DESIGN in the Abstract. At the moment this is not really evident until page 5 of the paper. The Conclusion could be strengthened. I feel that the time for just close monitoring of the rats of CS, especially in the private sector, has passed. The is now, as the author describe, considerable evidence that highlights the concerns related to over-medicalisation of childbirth and just monitoring is not in itself enough anymore. I recognise that this is a cross sectional analyses and therefore cannot determine cause and effect and can only look at associations, however, these are clear and consistent across the
---

	literature. I would like to see a much stringer statement about the need for action on the issues of incentives for CS in the paper. The authors discuss the imperative to have training for junior doctors on CS and how this might be a driver for higher rates in private facilities. Can the authors comment on the need for training in facilitating and support normal birth? In many ways this is harder than a CS as it requires patience, skills in observation and monitoring and a high level of care and support. We need to start valuing the skill in normal labour and birth rather than privileging the surgical skill in CS and in addition, incentivising normal labour and birth over CS. It would be helpful to have a comment to this effect in this paper. Finally, since the authors probably submitted this paper, The Lancet Series on Midwifery has been released (my conflict of interest is that I am a co-author). Paper 1 of this series (Renfrew et al, The Lancet, 2014) provides commentary and an analysis of the implications of over-medicalisation of childbirth, in particular, a high CS rate. It would be beneficial if the authors of this paper referred to The Lancet series as the two pieces of evidence are highly supportive of one another.
--	--

REVIEWER	Hmwe Hmwe Kyu Institute for Health Metrics and Evaluation, University of Washington, United States
REVIEW RETURNED	07-Aug-2014

GENERAL COMMENTS	The way the descriptive results (i.e., proportions of caesarean births) are presented in the abstract is confusing because there is no contrast between private and public facilities. For example, by only saying that 30% of caesarean deliveries were private facility births in rural Nepal, one might wrongly assume that the remaining 70% were public facility births while this is not true. It would be easier for the reader to follow if the corresponding proportions of caesarean births at public facilities are also presented for each location. Limitations should be discussed more thoroughly. The authors mentioned that their data were not nationally representative. They should also acknowledge the fact that the data were not sub-nationally representative. For example, Mumbai slums are not representative of India's urban population. Moreover, they used data from the control areas of the trials only. How could this influence the generalizability of the results? This should also be discussed. The way the analysis was conducted to test the interaction between maternal education and private facility delivery as shown in table 4 seems incorrect. Actually, interactions are multiplicative rather than additive. By doing the analysis that way, one cannot say whether there really was a significant interaction. Interaction terms should be the products of variables in the regression model. For example, in this study, the private facility variable has two categories, coded as 0 for public and 1 for private while the education variable has four categories which are represented by three dummy/indicator variables. Three interaction terms could be created using these variables: private facility*primary education; private facility*secondary education; and private facility*bachelor's degree. If one or more of these interaction terms are statistically significant,
--

	one could then say that an interaction exists between maternal education and private facility delivery. The estimates for these three interaction terms could be presented together with the main effects in table 3. Table 4 is actually not necessary. Instead of using the term "urban India" throughout the paper, more precise terms (e.g. "urban slums in India" or "Mumbai slums in urban India") should be used. Page 11, lines 39-40: The sentence referring to "the requirement for junior doctors to practice their surgical skills" needs a reference. Page 11, lines 44-48: It is not clear why well-educated women "act as training centers for junior doctors". This needs an explanation and a reference. Page 11, lines 50-54: The sentence "16% of facility births were by caesarean (in rural Nepal), which is higher than the national average of 5%" is misleading because the denominator constitutes only institutional births in the former while it includes both institutional and non-institutional deliveries in the latter. If both types of deliveries were included in the denominator for rural Nepal, the proportion of caesarean births would be about 5% (249/4931) instead of 16% (249/1586).
--	--

VERSION 1 – AUTHOR RESPONSE

Reviewer 1:

1. The fact that this is a secondary analysis of data from previous trials needs to be included under DESIGN in the Abstract.

We have specified that the study is a secondary, cross-sectional analysis in the Design section of the abstract.

2. The Conclusion could be strengthened. I feel that the time for just close monitoring of the rates of CS, especially in the private sector, has passed. There is now, as the author describe, considerable evidence that highlights the concerns related to over-medicalisation of childbirth and just monitoring is not in itself enough anymore. I recognise that this is a cross sectional analysis and therefore cannot determine cause and effect and can only look at associations, however, these are clear and consistent across the literature. I would like to see a much stringer statement about the need for action on the issues of incentives for CS in the paper.

We have rephrased the conclusion as follows:

“These findings urgently call for greater monitoring of the role of private providers in maternity care, together with action to mitigate the negative consequences of increased caesarean sections in the private sector for maternal and neonatal health outcomes. It remains the explicit responsibility of the medical profession and associated regulatory bodies to ensure that surgical intervention in delivery care is undertaken exclusively in the interests of the mother and unborn child, and not those of care providers.”

3. The authors discuss the imperative to have training for junior doctors on CS and how this might be a driver for higher rates in private facilities. Can the authors comment on the need for training in facilitating and support normal birth? In many ways this is harder than a CS as it requires patience, skills in observation and monitoring and a high level of care and support. We need to start valuing the

skill in normal labour and birth rather than privileging the surgical skill in CS and in addition, incentivising normal labour and birth over CS. It would be helpful to have a comment to this effect in this paper.

The reference to the imperative to train junior doctors to perform caesarean sections was added after co-authors from Bangladesh highlighted this as a possible driver for increased caesarean section rates. We have clarified that this is a hypothesis based on local experience in the following sentence: "Our own experience also suggests that such facilities have additional incentives to encourage caesareans in order to allow junior doctors to practice their surgical skills."

4. Finally, since the authors probably submitted this paper, The Lancet Series on Midwifery has been released (my conflict of interest is that I am a co-author). Paper 1 of this series (Renfrew et al, The Lancet, 2014) provides commentary and an analysis of the implications of over-medicalisation of childbirth, in particular, a high CS rate. It would be beneficial if the authors of this paper referred to The Lancet series as the two pieces of evidence are highly supportive of one another.

We thank the author for this reference and have highlighted it in the conclusion of the article: Data recently reviewed in the Lancet Midwifery Series suggest that even in countries with a rapid growth in caesarean sections, such as Brazil and China, steps can be taken to correct an over-reliance on obstetric care through the enhancement of midwifery-led services, and that these initiatives could help to curb caesarean section rates. 47 Exploring the revival of midwifery-led services in South Asia is an important future direction for research and action. 48

Reviewer 2

1. The way the descriptive results (i.e., proportions of caesarean births) are presented in the abstract is confusing because there is no contrast between private and public facilities. For example, by only saying that 30% of caesarean deliveries were private facility births in rural Nepal, one might wrongly assume that the remaining 70% were public facility births while this is not true. It would be easier for the reader to follow if the corresponding proportions of caesarean births at public facilities are also presented for each location.

We have corrected the abstract to:

The proportion of births by caesarean section in private versus public facilities was 73% vs. 29% in Bangladesh, 30% vs. 14% in rural Nepal, 18% vs. 15% in informal settlements of urban India, and 5% vs. 15% in rural India.

2. Limitations should be discussed more thoroughly. The authors mentioned that their data were not nationally representative. They should also acknowledge the fact that the data were not sub-nationally representative. For example, Mumbai slums are not representative of India's urban population. Moreover, they used data from the control areas of the trials only. How could this influence the generalizability of the results? This should also be discussed.

We have now acknowledged that the data are not sub-nationally representative in the limitations summary (p.3) and in the discussion (p.10). We did not discuss the use of control area data further: in all trials, randomisation ensured that the participants' characteristics were broadly comparable between intervention and control arms, as one would expect.

3. The way the analysis was conducted to test the interaction between maternal education and private facility delivery as shown in table 4 seems incorrect. Actually, interactions are multiplicative rather than additive. By doing the analysis that way, one cannot say whether there really was a significant interaction. Interaction terms should be the products of variables in the regression model. For example, in this study, the private facility variable has two categories, coded as 0 for public and 1 for private while the education variable has four categories which are represented by three dummy/indicator variables. Three interaction terms could be created using these variables: private facility*primary education; private facility*secondary education; and private facility*bachelor's degree. If one or more of these interaction terms are statistically significant, one could then say that an interaction exists between maternal education and private facility delivery. The estimates for these

three interaction terms could be presented together with the main effects in table 3. Table 4 is actually not necessary.

The reviewer questioned the modelling technique used to estimate how educational attainment modifies the association between type of facility and caesarean delivery. Below, we clarify how the method used in our analysis, which constructs indicator variables for each facility by education subgroup, is equivalent to the method suggested by the reviewer. We also explain why we chose to present results using this method to parameterize effect modification, and clarify how statistical significance of this effect modification was tested.

The parameterization of effect modification presented by the reviewer can be written as follows:

Where α_0 is the log odds of the outcome (in our example, caesarean delivery) among the reference group (uneducated women delivering in public facilities), α_1 is the main association of delivering in a private facility among uneducated women, α_2 is the main association of having primary education (PE) among those delivering in a public facility, α_3 is the main association of having secondary education (SE) among those delivering in a public facility, α_4 is the differential association of delivering in a private facility (PRI) compared to public among women with primary education, and α_5 is the differential association of delivering in a private facility compared to public among women with secondary education. (XA is a vector of additional covariates and associated parameters.)

For most policy and operational purposes, these main effects and differential associations are not particularly useful. Rather, we are interested in comparing associations between different socio-demographic factors and caesarean delivery with a single reference group, and in understanding whether these associations differ significantly from the null. For example, in our study we may be interested in understanding how different the odds of caesarean are among the most privileged (highly educated, delivering in private facility) compared with the least (no education, public facility delivery). In this case, we are interested in interpreting $(\alpha_3 + \alpha_5)$ and its confidence interval, rather than the separate parameters and confidence intervals for α_3 and α_5 .

The model presented in this paper is algebraically identical to the model suggested by the reviewer:

β_5 in the second model is equal to $(\alpha_3 + \alpha_5)$ in the first: the log odds ratio comparing caesarean delivery among highly educated women attending private facilities compared with uneducated women (UE) delivering in public facilities (PUB). The other parameters in this second model are also algebraic combinations of those estimated in the first: $\beta_1 = \alpha_1$, $\beta_2 = \alpha_2$, $\beta_3 = (\alpha_2 + \alpha_4)$, and $\beta_4 = \alpha_3$.

Table 4 of our analysis presents odds ratios estimated using the second parameterization of the effect modification presented above, the confidence intervals for these odds ratios, and the number of respondents included in each education by facility subgroup. We prefer this method of presenting these results for two reasons. The primary benefit is in the ease of interpreting the parameters and their associated confidence intervals. Comparing the odds of caesarean in each subgroup relative to a single reference allows for easier understanding of the differences in the odds of caesarean for each group, and facilitates comparisons across all groups. Confidence intervals are constructed by group, facilitating quick identification of the size and significance of each association. Second, presenting the count of women within each subgroup provides the reader with additional information about the data underlying the parameter estimate; we believe that this is an important addition to our results presentation. (More detail on parameterizing, reporting, and interpreting effect modification and the importance of parameters estimating differences from a single reference group can be found in Knol and Vanderweele (2012).

Finally, we used a likelihood ratio test to estimate whether the fit of the model improved with the additional modelling of effect modification compared with a model including main effects of place of delivery and educational attainment only. This method jointly tests the significance of multiple

parameters, rather than testing the significance of each parameter individually using a Wald test, as suggested by the reviewer.

4. Instead of using the term "urban India" throughout the paper, more precise terms (e.g. "urban slums in India" or "Mumbai slums in urban India") should be used.

We agree with this reviewer and have now replaced 'urban India' with 'informal settlements in urban India' throughout the manuscript.

5. Page 11, lines 39-40: The sentence referring to "the requirement for junior doctors to practice their surgical skills" needs a reference. Page 11, lines 44-48: It is not clear why well-educated women "act as training centers for junior doctors". This needs an explanation and a reference.

As explained above, the reference to the imperative to train junior doctors to perform caesarean sections was added after co-authors from Bangladesh highlighted this as a possible driver for increased caesarean section rates. We have now clarified that this is a hypothesis based on local experience on p.11.

6. Page 11, lines 50-54: The sentence "16% of facility births were by caesarean (in rural Nepal), which is higher than the national average of 5%" is misleading because the denominator constitutes only institutional births in the former while it includes both institutional and non-institutional deliveries in the latter. If both types of deliveries were included in the denominator for rural Nepal, the proportion of caesarean births would be about 5% (249/4931) instead of 16% (249/1586).

The reviewer is absolutely correct and we have rectified this error on p.11:

In our sample of institutional deliveries from rural Nepal, 5% of all births were by caesarean, which is similar to than the national average: the 2011 Nepal DHS found an overall national caesarean section rate of 4.6% [...]

Reference:

Knol MJ and VanderWeele TJ. Recommendations for presenting analyses of effect modification and interaction. *Int. J. Epidemiol.* (2012) 41 (2): 514-520. doi: 10.1093/ije/dyr218

VERSION 2 – REVIEW

REVIEWER	Caroline Homer University of Technology Sydney, Australia
REVIEW RETURNED	13-Sep-2014

GENERAL COMMENTS	The authors have addressed my previous comments adequately.
---

REVIEWER	Hmwe Hmwe Kyu Institute for Health Metrics and Evaluation, University of Washington
REVIEW RETURNED	21-Sep-2014

GENERAL COMMENTS	I appreciate the author's response to my comments. Although the revised manuscript has improved, there are a couple of important concerns that remain inadequately addressed. There is no adequate information provided to draw conclusions about the "size and statistical significance" of interactions as recommended in STROBE (Vandenbroucke et al., 2007) or the paper by Knol and Vanderweele (2012). For example, the recommendations by Knol and Vanderweele (2012) includes presenting: (i) odds ratios (ORs) for
---

each (public/private, education) stratum with a single reference category; (ii) ORs for public/private within strata of education and for education within strata of public/private; (iii) interaction measures on additive and multiplicative scales. The authors presented the first step [i.e., ORs for each (public/private, education) stratum with a single reference category] in their manuscript (Table 4) but omitted steps (ii) and (iii). Without (iii) (i.e., interaction measures on additive and multiplicative scales), one cannot know if an interaction or effect modification really exists. The authors responded to my comment to test and include interaction measures by providing two equations and stated that they are the same and that the second equation has been applied. In fact, they are different: the first equation estimates interactions on multiplicative scales (step (iii) in the above example) while the second equation represents step (i) in the above example. The likelihood ratio test that the authors has conducted just compares the fit of two models: it does not provide information about the size and statistical significance of interactions. The presence of an interaction implies that the joint effect of two factors are greater than the sum of the estimated effects of each factor alone. For example, the estimated joint effect of "private facility and primary education" together would be greater than the sum of the estimated effects of "private facility-no education" alone and "public facility-primary education" alone if there is an interaction. It would be very informative if the authors could use the template recommended by Knol and Vanderweele (2012) and report all steps. If they could not, they should at least report one of the following interaction measures as recommended in STROBE (Vandenbroucke et al., 2007): (i) Relative excess risk due to interaction (RERI) with confidence intervals (CIs); (ii) measure of interaction on a multiplicative scale with CIs.

The research objective on pages 4 & 5 ("to examine interactions between maternal education and caesarean section delivery in private and public facilities") implies examining the interaction between exposure (maternal education) and outcome (caesarean section delivery), which is very confusing. Isn't it examining the interaction between two exposures (i.e., maternal education and private/public facility) on the outcome (caesarean section delivery)? If so, it should be stated as such. The key message which states "We found significant interactive associations between maternal education and caesarean delivery in Bangladesh (both private and public facilities) and in informal Indian urbansettlements (private facilities only)" is confusing in a similar way and should be rephrased clearly. Finally, the following statement in the abstract "We found significant interactive associations between maternal education and private facility delivery in two of four locations (p=0.025 in rural Bangladesh and p<0.001 in informal settlements in urban India), with highly educated women particularly likely to deliver by caesarean in private facilities" could be misleading: the p-values are actually from the likelihood ratio test to test model fit but one could wrongly assume that they are the p-values for the interactions. To avoid this ambiguity, interaction measures with CIs and p-values should be presented here instead.

References

Knol MJ and VanderWeele TJ. Recommendations for presenting analyses of effect modification and interaction. *Int. J. Epidemiol.* 2012; 41 (2): 514-520. doi: 10.1093/ije/dyr218

	Vandenbroucke JP, Von Elm E, Altman DG, Gøtzsche PC, Mulrow CD, Pocock SJ, et al. Strengthening the reporting of observational studies in epidemiology (STROBE): explanation and elaboration. PLoS Medicine. 2007;4:e297. doi: 10.1371/journal.pmed.0040297
--	---

VERSION 2 – AUTHOR RESPONSE

Response to comments, 24 October 2014

Thank you for the opportunity to respond to comments and revise this manuscript prior to publication. Our response to the comments from reviewer 2 is below.

I appreciate the author's response to my comments. Although the revised manuscript has improved, there are a couple of important concerns that remain inadequately addressed. There is no adequate information provided to draw conclusions about the "size and statistical significance" of interactions as recommended in STROBE (Vandenbroucke et al., 2007) or the paper by Knol and Vanderweele (2012). For example, the recommendations by Knol and Vanderweele (2012) includes presenting: (i) odds ratios (ORs) for each (public/private, education) stratum with a single reference category; (ii) ORs for public/private within strata of education and for education within strata of public/private; (iii) interaction measures on additive and multiplicative scales. The authors presented the first step [i.e., ORs for each (public/private, education) stratum with a single reference category] in their manuscript (Table 4) but omitted steps (ii) and (iii). Without (iii) (i.e., interaction measures on additive and multiplicative scales), one cannot know if an interaction or effect modification really exists.

The authors responded to my comment to test and include interaction measures by providing two equations and stated that they are the same and that the second equation has been applied. In fact, they are different: the first equation estimates interactions on multiplicative scales (step (iii) in the above example) while the second equation represents step (i) in the above example. The likelihood ratio test that the authors has conducted just compares the fit of two models: it does not provide information about the size and statistical significance of interactions. The presence of an interaction implies that the joint effect of two factors are greater than the sum of the estimated effects of each factor alone. For example, the estimated joint effect of "private facility and primary education" together would be greater than the sum of the estimated effects of "private facility-no education" alone and "public facility-primary education" alone if there is an interaction. It would be very informative if the authors could use the template recommended by Knol and Vanderweele (2012) and report all steps. If they could not, they should at least report one of the following interaction measures as recommended in STROBE (Vandenbroucke et al., 2007): (i) Relative excess risk due to interaction (RERI) with confidence intervals (CIs); (ii) measure of interaction on a multiplicative scale with CIs.

We have provided stratum-specific interaction measures (i.e., differential associations between caesarean and private delivery within educational strata) for each location as supplemental table 2. We believe that our presentation of results fits within the STROBE criteria for appropriate presentation of interactions, and that adding additional calculations, such as prevalence differences, would not substantially improve the reader's understanding of the determinants of caesarean delivery. Moreover, we think that the most intuitive presentation of the interaction, comparing all subgroups to a single, meaningful reference group, is included and described clearly in the manuscript.

We have also reworded the results section (page 9) to clearly represent the likelihood ratio test as a test of model fit, not individual parameters.

The research objective on pages 4 & 5 ("to examine interactions between maternal education and

caesarean section delivery in private and public facilities") implies examining the interaction between exposure (maternal education) and outcome (caesarean section delivery), which is very confusing. Isn't it examining the interaction between two exposures (i.e., maternal education and private/public facility) on the outcome (caesarean section delivery)? If so, it should be stated as such.

We have reworded this statement to clarify that the interaction is between place of delivery and mother's educational attainment (page 5)

The key message which states "We found significant interactive associations between maternal education and caesarean delivery in Bangladesh (both private and public facilities) and in informal Indian urban settlements (private facilities only)" is confusing in a similar way and should be rephrased clearly.

We have clarified this text (page 3).

Finally, the following statement in the abstract "We found significant interactive associations between maternal education and private facility delivery in two of four locations ($p=0.025$ in rural Bangladesh and $p<0.001$ in informal settlements in urban India), with highly educated women particularly likely to deliver by caesarean in private facilities" could be misleading: the p-values are actually from the likelihood ratio test to test model fit but one could wrongly assume that they are the p-values for the interactions. To avoid this ambiguity, interaction measures with CIs and p-values should be presented here instead.

We have included the odds ratios and 95% CIs in the abstract (page 2).

VERSION 3 - REVIEW

REVIEWER	Hmwe Hmwe Kyu Institute for Health Metrics and Evaluation, University of Washington
REVIEW RETURNED	12-Nov-2014

GENERAL COMMENTS	The results in table 4 and supplementary table 2 are consistent for urban India but are completely contradictory to each other for rural Bangladesh. Table 4 shows that women with higher education delivered at private facilities were more likely to have cesarean deliveries but supplementary table 2 indicates that they are less likely to have the procedure in rural Bangladesh. This discrepancy needs to be explained.
---

VERSION 3 – AUTHOR RESPONSE

Response: Thank you for your additional comments and queries. The results given in table 4 and supplementary table 2 are not contradictory; rather, they highlight different disparities within the population of Bangladeshi women included in our sample.

The results in table 4 show the difference in caesarean prevalence between the most educated women in private facilities and the least educated in public facilities; we highlight this substantial difference as an example of the diverse experiences of care received in an underserved, rural and low-income population. The supplementary table includes estimates of the difference between more and less educated women in their difference in prevalence of delivery in public and private facilities. Both the main associations of educational attainment and private delivery facility are very high in Bangladesh, but the combined association of educational attainment and private delivery is somewhat

less high than would be expected given the individual main effects.

We have added a sentence to the paragraph presenting the interaction association results (paragraph 1 on page 10) noting the negative interaction association in Bangladesh, and have modified the abstract (page 2), article summary (page 3), introduction (page 4), and discussion (page 12).